# Effect of Hyperbaric Oxygen and Inflammation on Human Gingival Mesenchymal Stem/Progenitor Cells

**DOI:** 10.3390/cells12202479

**Published:** 2023-10-18

**Authors:** Johannes Tölle, Andreas Koch, Kristina Schlicht, Dirk Finger, Wataru Kaehler, Marc Höppner, Christian Graetz, Christof Dörfer, Dominik M. Schulte, Karim Fawzy El-Sayed

**Affiliations:** 1Clinic for Conservative Dentistry and Periodontology, School of Dental Medicine, Christian-Albrechts-University, 24105 Kiel, Germany; toelle.johannes@t-online.de (J.T.); dirk.finger@mailbox.org (D.F.); graetz@konspar.uni-kiel.de (C.G.); doerfer@konspar.uni-kiel.de (C.D.); 2German Naval Medical Institute, 24119 Kiel, Germany; a.koch@iem.uni-kiel.de (A.K.); w.kaehler@iem.uni-kiel.de (W.K.); 3Institute of Diabetes and Clinical Metabolic Research, University Hospital Schleswig-Holstein, 24105 Kiel, Germany; kristina.schlicht@uksh.de (K.S.); dominikmaria.schulte@uksh.de (D.M.S.); 4Institute of Clinical Molecular Biology, School of Medicine, Christian-Albrechts-University, 24105 Kiel, Germany; m.hoeppner@ikmb.uni-kiel.de; 5Division of Endocrinology, Diabetes and Clinical Nutrition, Department of Internal Medicine I, University Hospital Schleswig-Holstein, 24105 Kiel, Germany; 6Oral Medicine and Periodontology Department, Faculty of Dentistry, Cairo University, Cairo 12613, Egypt

**Keywords:** inflammation, stem cells, hyperbaric oxygen, regeneration, periodontitis, ROS, next-generation sequencing

## Abstract

The present study explores for the first time the effect of hyperbaric oxygen (HBO) on gingival mesenchymal stem cells’ (G-MSCs) gene expression profile, intracellular pathway activation, pluripotency, and differentiation potential under an experimental inflammatory setup. G-MSCs were isolated from five healthy individuals (*n* = 5) and characterized. Single (24 h) or double (72 h) HBO stimulation (100% O2, 3 bar, 90 min) was performed under experimental inflammatory [IL-1β (1 ng/mL)/TNF-α (10 ng/mL)/IFN-γ (100 ng/mL)] and non-inflammatory micro-environment. Next Generation Sequencing and KEGG pathway enrichment analysis, G-MSCs’ pluripotency gene expression, Wnt-/β-catenin pathway activation, proliferation, colony formation, and differentiation were investigated. G-MSCs demonstrated all mesenchymal stem/progenitor cells’ characteristics. The beneficial effect of a single HBO stimulation was evident, with anti-inflammatory effects and induction of differentiation (*TLL1*, *ID3*, *BHLHE40*), proliferation/cell survival (*BMF*, *ID3*, *TXNIP*, *PDK4*, *ABL2*), migration (*ABL2*) and osteogenic differentiation (*p* < 0.05). A second HBO stimulation at 72 h had a detrimental effect, significantly increasing the inflammation-induced cellular stress and ROS accumulation through *HMOX1*, *BHLHE40*, and *ARL4C* amplification and pathway enrichment (*p* < 0.05). Results outline a positive short-term single HBO anti-inflammatory, regenerative, and differentiation stimulatory effect on G-MSCs. A second (72 h) stimulation is detrimental to the same properties. The current results could open new perspectives in the clinical application of short-termed HBO induction in G-MSCs-mediated periodontal reparative/regenerative mechanisms.

## 1. Introduction

Periodontitis is a multifactorial inflammatory disorder associated with microbial dysbiosis [1]. The initial localized inflammatory reaction is characterized by an increased expression of the pro-inflammatory cytokines interleukin-1β (IL-1β), interleukin-4 (IL-4), interleukin-6 (IL-6), and tumor necrosis factor-α (TNF-α) [2]. On prolonged periodontal inflammation, further expression of interferon-γ (INF-γ) results in additional cellular stress, osteoclast activation, bone resorption, and tissue destruction [3,4,5]. Aside from the regenerative potential of periodontal mesenchymal stem/progenitor cells (MSCs), MSCs possess the ability to modulate such inflammatory responses via NF-κB- and Wnt-/β-catenin-pathway-dependent manners [6,7,8]. Yet, with increased periodontal disease progression, such MSCs’ reparative/regenerative potential would substantially be hampered by rising intracellular levels of reactive oxygen species (ROS) with subsequent caspase-3-dependent cellular apoptosis [1,5,9,10,11,12].

Short-termed hyperbaric oxygen (HBO; 100% pressurized oxygen (≥1 atm)) [13,14], therapeutically employed in non-healing chronic injuries, is believed to increase tissue oxygenation and ROS elimination, through elevation of the superoxide dismutase expression (SOD2), the antioxidant protein that catalyzes the mitochondrial ROS dismutation into oxygen and hydrogen peroxide [15,16,17]. HBO further reduced TNF-α, IL-6, and IL-10 expression and enhanced survival in a sepsis model [18]. HBO stimulation was reported to activate the Wnt-/β-Catenin pathway in adipose-derived stem cells, Wharton jelly mesenchymal stem cells, and neural stem cells [19,20,21] and to improve MSCs proliferation and differentiation [22,23,24]. Earlier results further suggested a positive impact of HBO-therapy on clinical attachment levels, bleeding scores, and microbial compositions in periodontitis patients [25,26]. Thus, the current study explored for the first time the effect of a single (24 h) or double (72 h) short-termed HBO therapy on G-MSCs’ mRNA expression and intracellular pathway activation via Next Generation Sequencing (NGS) and KEGG pathway analysis, pluripotency gene expression, Wnt-/β-catenin pathway activation, proliferation, colony-formation and differentiation under experimental inflammatory and non-inflammatory conditions.

## 2. Materials and Methods

### 2.1. Cell Culturing

The study’s protocol was approved by the ethical committee of the Christian Albrechts University of Kiel, Kiel, Germany (IRB-Approval D513/17). After obtaining the patients’ informed consents, retromolar gingiva samples were surgically obtained from five randomly recruited healthy individuals (*n* = 5; Appendix A), during elective interventions. Cells were isolated, cultured and STRO-1 immunomagnetically sorted (Table 1). Subsequently, cultures of STRO-1+ sorted G-MSCs [27] were purified by single colony selection.

### 2.2. Experimental Groups

Third passage G-MSCs were preconditioned in the following four groups and received a single treatment at 24 h or a double treatment over 72 h (US-Navy treatment protocol Table 9) [28].

### 2.3. Control Group: Basic Medium

HBO-group: basic medium supplemented with hyperbaric oxygen (100% O2, [Hüllmann Gas, Hans Hüllmann GmbH, Kiel, Germany], 90 min, 2–3 bar).

Inflammatory group: basic medium supplemented with inflammatory cytokines (IL-1β [1 ng/mL], TNF-α [10 ng/mL], and IFN-γ [100 ng/mL]) [29,30,31,32,33,34].

HBO/inflammatory group: basic medium with hyperbaric oxygen (100% O2, [Hüllmann Gas], 90 min, 2–3 bar) and inflammatory cytokines (IL-1β [1 ng/mL], TNF-α [10 ng/mL] and IFN-γ [100 ng/mL]).

Media were renewed every second day. To create hyperbaric conditions, samples were placed in an HBO-chamber system (Testcom200/20, Haux-Life-Support, Karlsbad-Ittersbach, Germany), nitrogen layers were exhausted, by applying pure O2, and the pressure of 2–3 bar was built-up, following an established protocol [31] composed of compression (10 min), supplementation of isobaric oxygen (90 min) and finally decompression (10 min). Pneumatic use of pure oxygen is prohibited due to its high reactivity (IGV-SH-07S-Rev1). Instead, samples were placed in a compressed air chamber and received constant oxygen ventilation through individually constructed covers (Appendix A).

### 2.4. Flow Cytometric Analysis

Third passage G-MSCs were characterized for their surface marker expression (CD14, CD34, CD45, CD73, CD90, CD105) [35,36,37]. Cells were bound to fluorochrome-linked antibodies and measured against their corresponding isotype controls (FACSCalibur E6370 and FACSComp 5.1.1 software [BD Biosciences, Franklin Lakes, NJ, USA]). Background noise was reduced with an FcR-blocking reagent (Miltenyi Biotec) according to the manufacturers’ instructions.

### 2.5. Real-Time-Polymerase-Chain-Reaction

In the third passage G-MSCs, *NANOG* (NANOG homeobox protein), *OCT4A* (octamer-binding transcription factor 4A; Pou5f1) and *SOX2* (sex determining region Y-box containing gene 2) mRNA expression was measured via real-time quantitative polymerase chain reaction (RT-qPCR). Reaching 80% confluence, mRNA extraction and cDNA synthesis were performed using Rneasy^®^ Protect Mini Kit (250) and QuantiTect^®^ Reverse Transcription Kit (Qiagen, Hilden, Germany). Each sample received a 20 µL reaction mixture [4 pmol primer, 10 μL LightCycler Probes Master mixture (Roche Life Science, Prenzberg, Germany), and 5 μL specimen cDNA]. RT-qPCR was run for 40 cycles (Light Cycler 96; Roche). All trials were performed thrice and averaged. mRNA expression was normalized to *PGK1* (phosphoglycerate kinase 1) housekeeping gene. *PGK1* was identified as the most suitable reference of 19 reference genes via NormFinder algorithm (v.05/01-2015; Department of Molecular Medicine; Aarhus University Hospital, Denmark). Relative quantification of gene expression levels was calculated via the 2^−ΔΔCt^ method.

### 2.6. Next Generation Sequencing

The setup described under “Experimental groups” allows a variety of possible comparisons and modeling methods with regard to differential RNA expression. In total, RNA was extracted from 28 samples from 4 probands. However, due to sequencing failures in some samples, to keep this analysis as concise as possible and in response to the dominating effects of inflammation conditions, the main focus of NGS analyses was put on differential expression analysis (DEA) in HBO/inflammation samples compared to the inflammation samples directly, as well as on HBO treatment under normal (non-inflammation) conditions. For this, 16 paired samples taken from two different probands were used. DEA was done after single (24 h) and double (72 h) HBO stimulation. Next Generation Sequencing was performed at the NGS lab at the Institute of Clinical Molecular Biology (IKMB) in Kiel (Illumina MiSeq; San Diego, CA, USA). Raw FastQ files were aligned, quality controlled, and transformed into read counts using the Nextflow nfcore/RNAseq pipeline [35]. Read counts were analyzed in R v3.6.2 via edgeR and DeSEQ2 packages. Gene counts were rlog transformed and plotted in heatmaps in DeSEQ2. Differential expression analysis (DEA) was done in edgeR using the Quasi-Likelihood F-test function with multi-level contrasting. Contrasts in the main analysis were modeled for the effect of HBO under inflammation and normal conditions at 24 h and 72 h. Appendix A provides an overview of additional contrasts for the effect of inflammatory conditions compared to normal conditions without HBO treatment. In the correction of false discovery rates (FDR), the Benjamin–Hochberg method was used. Pathway enrichment analysis was done for all differentially expressed genes using the Kyoto Encyclopedia of Genes and Genomes (KEGG) database and visualized using the R package “clusterprofiler”.

### 2.7. ELISA pS45/Total β-Catenin

Phospho-β-catenin 45 (pS45) and total β-catenin were measured following a single HBO stimulation (24 h) in each group. ELISA was performed using ab205705β-Catenin (pS45) and total β-catenin SimpleStep ELISA^®^Kit (Abcam, Cambridge, UK), following the manufacturer’s instructions. 2 × 104 G-MSCs/well were seeded out on 24-well-palates. After a single stimulation at 24 h, samples were washed twice (PBS [Biochrom]) and lysated via 1× Cell Extraction Buffer PTR (100–500 µg/mL of protein). 50 µL of each sample were transferred to the antibody-coated 96-well-plate and primary and secondary antibodies (β-Catenin (Total) Capture Antibody/β-Catenin (Total) Detector Antibody and β-Catenin (pS45) Capture Antibody/b-Catenin (pS45) Detector Antibody) were added. Redundant antibodies were removed by washing four times with 350 µL 1× Wash Buffer PT one hour later. After pitching the supernatants, 100 µL of TMB Substrate was added followed by incubation (14 min, shaking 400 rpm, in the dark). Finally, the reaction was stopped by adding 100 µL Stop Solution. Optical density was measured and compared to standard curves (λ = 450 nm; MultiscanGo, Thermo Fisher Scientific, Waltham, MA, USA).

### 2.8. Cellular Proliferation and Colony-Forming Units (CFUs)

Cellular proliferation was examined over 12 days. 1 × 10^3^ third passage G-MSCs were seeded on 24-well-hard-shell plates and attached for 24 h. One well of each group was counted every 24 h (Neubauer chamber, Roth, Karlsruhe, Germany).

1 × 102 third passage G-MSCs (1.65 cells/cm^2^) were seeded out on cell culture dishes and grew for 12 days. Colonies were fixed with 80% ethanol (Sigma-Aldrich, St. Louis, MO, USA) and stained (crystal violet, Merck, Darmstadt, Germany). Clusters consisting of a minimum of 50 cells were counted as colonies. Three independent investigators counted colonies and the arithmetical averages were calculated and statistically evaluated.

### 2.9. Multilinear Differentiation

Following a single stimulation in the respective groups, G-MSCs were examined for their multilinear differentiation potential into osteogenic, adipogenic, and chondrogenic directions as described before [38].

### 2.10. Statistical Analysis

Data was checked for normality, using the Shapiro-Wilk-test and tested for homogeneity of variance by Levene testing and graphical exploration. Significances between groups regarding gene expression, β-catenin (total/pS45) activation, colony formation, and multilinear differentiation were tested via Friedman non-parametric ANOVA. Post-hoc, pairwise comparisons were performed via Wilcoxon-signed-rank-test. The level of significance was set as *p* ≤ 0.05. Multiple testing was adjusted via the Bonferroni method. Data transformation and testing were calculated with MS Excel (Microsoft, Washington, DC, USA, v.16.0) and SPSS (IBM, Chicago, IL, USA, v.18.0.0).

### 2.11. Data Availability

FastQ files were uploaded to the Gene Expression Omnibus (GEO) platform and can be accessed publicly (GSE244404). Further, all data will be available from the corresponding author upon reasonable request.

## 3. Results

### 3.1. G-MSCs’ Characterization

Cells grew out of their soft tissue masses (Figure 1A) and demonstrated fibroblast-like condensed colonies (Figure 1B). G-MSCs’ flowcytometric (Figure 1C) surface marker profiling showed negative CD14, CD34, and minute CD45 expression, whilst high expression of CD73, CD90, and CD105. Following two weeks of osteogenic stimulation, cultures developed distinct calcification (Figure 1D) compared to their controls (Figure 1E). Three weeks after adipogenic induction, G-MSCs exhibited Red-Oil-O positive lipid droplet formation (Figure 1F) compared to their controls (Figure 1G). Chondrogenic differentiated cell masses (Figure 1H) demonstrated dense rather red-stained cortex (1) and loose blue-colored central (2) areas five weeks following stimulation, in contrast to their controls (Figure 1I) showing homogenous Nuclear-Fast-Red staining.

### 3.2. Next Generation Sequencing

DEA for HBO treatment under different conditions revealed 255 genes that were differentially regulated for the effect of HBO treatment under standard conditions after a single (24 h) stimulation, whereas the number of DE genes increased to 1999 with double HBO treatment after 72 h (Figure 2A,B). Likewise, for the effect of HBO treatment under inflammation conditions 134 genes were differentially expressed at 24 h, and 1536 genes following 72 h HBO stimulation (Figure 2C,D). The top three differentially expressed genes in both groups are shown below (Table 2); 24 h HBO treatment under standard conditions resulted in three overrepresented KEGG pathways: Human papillomavirus infection, calcium signaling pathway, and breast cancer (Figure 2C). After 72 h, we observed an activation of many genes in the Zinc-finger-protein family, with an underexpression of the Herpes simplex virus infection pathway. In contrast, genes in the BHLE family were overexpressed, resulting in the activation of the circadian rhythm pathway (Figure 2C). The breast cancer pathway was further activated through the *FGF18*, *HEYL*, and *WNT11* genes. Regarding 24 h HBO treatment under inflammation, the downregulation of several genes in the Inhibitor of the DNA binding family resulted in the activation of the TGF-β signaling pathway, regulating the pluripotency of stem cells and notch signaling (Figure 2D). With 72 h HBO treatment an activation of the circadian clock pathway through genes in the ZNF family, as well as the breast cancer pathway was observed (Figure 2D). For completeness, Appendix A provides results of additional contrasts for the effect of inflammatory conditions compared to normal conditions without HBO treatment at 24 h and 72 h. Comparison of inflammatory medium to standard medium revealed, as expected, a number of DE genes involved in inflammatory processes (e.g., *TNFAIP3*) and the activation of the cytokine–cytokine interaction KEGG pathway (Table 3).

### 3.3. Pluripotency Genes

*NANOG* expression after a single (24 h) stimulation (median gene expression/*PGK1*; Q25/Q75) was highest in inflammatory—(0.0002; 0.0001/0.0008), followed by HBO—(0.0001;0.0001/0.0012) with equally low levels in control—(0.0000; 0.0000/0.0002) and HBO/inflammatory-group (0.0000; 0.0000/0.0001; *p* > 0.05). Following a double (72 h) stimulation, *NANOG* was highest in inflammatory—(0.0003; 0.0001/0.0150) and control—(0.0002; 0.0001/0.0003), but nearly absent in HBO/inflammatory—(0.0000; 0.0000/0.0009) and HBO-group (0.0000; 0.0001/0.0003; *p* > 0.05; Figure 3A). *SOX2* expression after 24 h stimulation was highest in HBO—(0.0003; 0.0001/0.0045) followed by control—(0.0003; 0.0001/0.0007) and HBO/inflammatory—(0.0001; 0.0001/0.0002) with almost no expression in inflammatory-group (0.0000; 0.0000/0.0050). After 72 h, expression was highest in inflammatory—(0.0006; 0.0000/0.0011), followed by control—(0.0003; 0.0000/0.0007), HBO/inflammatory—(0.0001; 0.0000/0.0033) and HBO-group (0.0001; 0.0000/0.0023; *p* > 0.05; Figure 3B). *OCT4A* expression after 24 h was highest in HBO—(0.0007; 0.0002/0.0016) followed by control—(0.0002; 0.0001/0.0007), HBO/inflammatory—(0.0002; 0.0000/0.0007) and inflammatory-group (0.0001; 0.0000/0.0017). After 72 h, *OCT4A* was almost equally expressed in control—(0.0005; 0.0001/0.0008) and inflammatory—(0.0005; 0.0001/0.0007) as well as HBO—(0.0002; 0.0000/0.0005) and HBO/inflammatory-group (0.0001; 0.0000/0.0015; *p* > 0.05; Figure 3C).

### 3.4. ELISA for Total and pS45-β-Catenin

Total β-catenin levels (median concentration [µg/mL]; Q25/Q75) were highest in inflammatory—(497.00; 417.07/551.82), followed by HBO/inflammatory—(457.38; 389.20/554.59), control—(447.08; 413.85/472.13) and HBO-group (415.92; 362.21/496.54; *p* > 0.05; Figure 4A). pS45 β-catenin was highest in HBO—(175.36; 131.72/182.88), followed by inflammatory—(170.03; 134.39/201.30), HBO/inflammatory—(168.57; 134.15/207.77) and control-group (162.27; 126.55/182.63; *p* > 0.05). Median pS45/total-β-catenin (Q25/Q75) was highest in HBO/inflammatory—(0.356; 0.329/0.401), followed by control—(0.353; 0.307/0.392), HBO—(0.346; 0.335/0.447) and inflammatory-group (0.336; 0.272/0.444; *p* > 0.05).

### 3.5. Cellular Proliferation and CFUs

Cell counts increased in all groups over 12 days. After 3 days, median cell numbers (Q25/Q75) were highest in control—(67500; 27500/68750), followed by inflammatory—(40000; 31250/102500), HBO/inflammatory (27500; 21250/53750) and HBO-group (25000; 18750/56250; *p* = 0.005; padjust = 0.02; Figure 4B). After 6 days, cells grew most in the control group (162500; 118750/167500) followed by a nearly equal growth in inflammatory—(122500; 98750/186250) and HBO group (122500; 100000/155000) and were least in HBO-inflammatory-group (110000; 73750/157500; *p* = 0.088; padjust = 0.352). After 9 days, counts were highest in inflammatory—(200000; 156250/213750), followed by control (192500; 153750/233750), HBO/inflammatory—(180000; 123750/202500) and HBO-group (170000; 131250/222500; *p* = 0.069; padjust = 0.276). Finally, after 12 days, cell count was highest in inflammatory—(237500; 185000/266250), followed by control—(227500; 180000/276250), HBO/inflammatory—(220000; 165000/243750) and HBO-group (217500; 167500/ 268750; *p* = 0.472; padjust > 1). Colony formation (median CFUs count; Q25/Q75) was highest in HBO—(73.67; 13.16/81), followed by control—(67.00; 14.00/77.00), inflammatory—(53.00; 14.33/108.16) and HBO/inflammatory-group (37.67; 9.83/78.17; *p* > 0.05; Figure 4C).

### 3.6. Multilinear Differentiation Potential

Osteogenic differentiation (relative Ca2+ content; Q25/Q75) was significantly higher in HBO—(0.201; 0.128/0.251) and HBO/inflammatory—(0.158; 0.096/0.222) compared to control—(0.016; 0.008/0.178) and inflammatory-group (0.013; 0.008/0.115; *p* < 0.05; Figure 5(A1)). Following adipogenic differentiation, lipid formation (relative lipid content; Q25/Q75) was highest in HBO—(0.027; 0.016/0.046) and HBO/inflammatory—(0.024; 0.018/0.045), followed by control—(0.023; 0.020/0.031) and finally inflammatory-group (0.020; 0.012/0.036; *p* > 0.05; Figure 5(B1)). Regarding chondrogenic differentiation, Alcian-Blue staining (relative pixel content Q25/Q75) was highest in HBO—(0.658; 0.576/0.704), closely followed by control—(0.641; 0.487/0.661), HBO/inflammatory—(0.625; 0.608/0.711) and finally inflammatory-group (0.549; 0.467/0.627; *p* > 0.05). Nuclear-Fast-Red staining was highest in inflammatory—(0.438; 0.361/0.518) followed by HBO/inflammatory—(0.357; 0.278/0.379), control—(0.347; 0.327/0.475) and HBO group (0.330; 0.286/0.411; *p* > 0.05; Figure 5(C1)).

Regarding differentiation gene expressions, following osteogenic differentiation and 24 h pre-stimulation, the expression of the *RUNX2* early osteogenic marker was highest in HBO/inflammatory—(0.1020; 0.0512/0.1430) and HBO—(0.0813; 0.0131/0.1580) but lower in inflammatory—(0.1020; 0.0521/0.1430) and control group (0.0356; 0.023/0.1487; *p* > 0.05; Figure 5(A3)). *ALP* was highest HBO/inflammatory—(0.0006; 0.0004/0.0019), followed by nearly equal expression in inflammatory—(0.0004; 0.0004/0.0008) and HBO—(0.0004; 0.0001/0.0029) and lowest expression in control (0.0003; 0.0001/0.0019; *p* > 0.05; Figure 5(A4)). In adipogenic differentiation and 24 h pre-stimulation, *PPARγ* expression was highest in inflammatory—(0.2700; 0.1050/0.3650), followed by control—(0.1200; 0.1200/0.6450), HBO—(0.0740; 0.0480/0.1500) and HBO/inflammatory-group (0.0680; 0.0210/0.1750; *p* > 0.05; Figure 5(B3)). *LPL* expression after 24 h pre-stimulation was nearly equal in inflammatory—(0.0170; 0.0145/0.0420) and HBO—(0.0170; 0.0091/0.0265), followed by control (0.0150; 0.0067/0.0505) and HBO/inflammatory-group (0.0100; 0.0039/0.0845; *p* > 0.05; Figure 5(B4)). *ACAN* expression after 24 h HBO/inflammation stimulation was highest in inflammatory—(3.2772; 0.8235/8.8987), followed by HBO/inflammatory—(1.5899; 0.4853/8.1237), control—(1.4651; 0.4623/2.423) and HBO-group (0.6404; 0.2040/4.2633; *p* < 0.05; Figure 5(C3)).

## 4. Discussion

G-MSCs represent a readily available minimally invasive source of stem/progenitor cells, with significant proliferation, differentiation, regeneration, and inflammation-modulatory attributes [7,11,38,39,40,41,42]. Similar to previous investigations, G-MSCs isolation was performed, employing the STRO-1 immunomagnetic separation method [27,35,43]. The isolated G-MSCs demonstrated all predefined MSCs’ characteristics, with remarkable CFUs, surface markers expression pattern, and multilineage differentiation aptitude [27,37]. The current study employed an experimental IL-1β/TNF-α/IFN-γ inflammatory cocktail to test the effect of a single (at 24 h) or double (at 72 h) HBO induction, with or without inflammatory stimuli, on the G-MSCs intracellular pathway activation, pluripotency gene expression, Wnt-/β-catenin pathway activation, proliferation, colony-formation and differentiation in vitro. The current investigation employed a short-termed 90 min HBO stimulation scheme, following the US Navy clinical protocol standard [28] and earlier clinical investigations [25,26], avoiding long-term HBO stimulation (i.e., continuous 24 h or 48 h 100% O2 stimulation) [44,45], which could result in oxygen intoxication.

Differential gene expression is by nature versatile. To describe the dominant alterations, we primarily focused on the top three HBO-induced DE genes. Interestingly, these genes strongly correlated with differentiation (*TLL1*, *ID3*, *BHLHE40*), proliferation/cell survival (*BMF*, *ID3*, *TXNIP*, *PDK4*, *ABL2*), and migration (ABL2). Similar to previous investigations [46], current NGS results revealed that HBO treatment counteracted inflammation-induced gene activation in G-MSCs. A single HBO stimulation activated the calcium signaling pathway, silencing the phosphatidylinositol signaling branch via *PTK2B*, *FGF18*, and *GRPR* inhibition [47], with a subsequent hampering of PIP3-dependent ROS synthesis, strengthening oxidative stress resistance [48] at 24 h. The anti-inflammatory effect of a single HBO application was further evident through the downregulation of *HMOX1*, primarily implicated in a multitude of inflammatory diseases (i.e., Crohn’s disease and ulcerative colitis). Further, downregulation of the *Wnt11* dependent branches in Brest Cancer and human papillomavirus infection pathways indicated reduced signaling within Wnt/β-Catenin signaling. Wnt/β-Catenin activation, however, is crucially associated with the early cytokine-induced inflammatory reaction and cell cycle progression [31]. *HEYL*, *FGF18*, *LFNG*, and *PHKG1* are part of *MAPK* [49] and Notch signaling branches, affecting the overall expression of the key inflammatory transcription factor NFκB [50] and proliferation/cell survival. The effect eventually affected the anti-inflammatory protection against an inflammatory stimulus after 24 h. Interestingly, the effect primarily targeted pro-inflammatory signaling (TGF-β/Notch signaling) and signaling pathways regulating pluripotency of stem cells but strongly enhanced the *BMPR1B*-associated differentiation potential [51].

Yet, with a second HBO treatment an opposite effect was evident at 72 h, with upregulated *HMOX1*, indicating an HBO/inflammation augmented ROS production [52], in addition to *FGF18*, *IGF1* knock-down in Breast Cancer signaling [53] and p53 linked *GADD45* and *HES1* activation, both signifying reduced cell survival and apoptosis [54]. Furthermore, following the second HBO application at 72 h, *PER2* activation in Circadian Rhythm signaling indicated ubiquitin-mediated proteolysis [55] degrading oxidized proteins, suggesting HBO-induced protein damage. The enrichment of the herpes simplex virus 1 infection pathway was created by a large suppression of different ZNF genes. *ZNFs* are discussed to be tumor suppressor genes and alterations are involved in inflammatory skin diseases and diabetes development [56,57]. As the beneficial effect of HBO on the intracellular survival and differentiation pathways was solely evident on a single stimulation, the subsequent multilineage experiments were conducted following a single short-termed HBO pre-stimulation.

Regarding cellular properties, an inflammation-induced enhanced cellular proliferation was evident in the G-MSCs, similar to previous study results [6,31,58,59]. This effect was counteracted by the anti-inflammatory effect of the HBO treatment, which downregulated intracellular pathways associated with proliferation and cell survival (*BMF*, *ID3*, *TXNIP*, *PDK4*). Yet, no differences were notable in the pluripotency *NANOG*/*OCT4A*/*SOX2* gene expression, Wnt-/β-Catenin signaling, or CFUs formation. The absence of Wnt-/β-Catenin signaling pathway activation, in contrast to earlier findings [19,21,60], could rely on the short-termed HBO stimulation protocol employed in the current study. A single HBO pre-stimulation downregulated *BMF* expression, induced via TNF-α in a RIP1-dependent mechanism [61,62], necessary for death-receptor-induced apoptosis/necroptosis [63], with improved G-MSCs’ survival and differentiation potential [64]. The single HBO pre-stimulation further downregulated the inflammation-induced *ID3* in G-MSCs. Such a decrease in *ID1-3* and increased *BMPR1B* expression would result in TGF-β signaling pathway activation and upregulation of pluripotency and differentiation potentials of G-MSCs [51,65], explaining the concurrently observed G-MSCs’ stronger osteogenic and chondrogenic differentiation upon a single HBO pre-stimulation [66].

## 5. Conclusions

Taken together, current results point to a positive short-term stimulatory effect of a single HBO induction on G-MSCs’ regenerative and differentiation properties relying primarily on its ability to hinder/downregulate ROS inflammation-associated apoptotic pathways, while inducing regenerative intracellular signaling pathways. Yet, a second stimulation within a short time (72 h) would augment the inflammatory stress inside the cells, occurring during periodontitis, with accelerated signs of cellular damage. Thus, a single short-term HBO induction could be considered a beneficial priming method of G-MSCs for periodontal regenerative approaches. Still, further exploration of the HBO ideal stimulation duration and frequency in addition to clinical interventional studies is required to confirm the observed effects.

## Figures and Tables

**Figure 1 cells-12-02479-f001:**
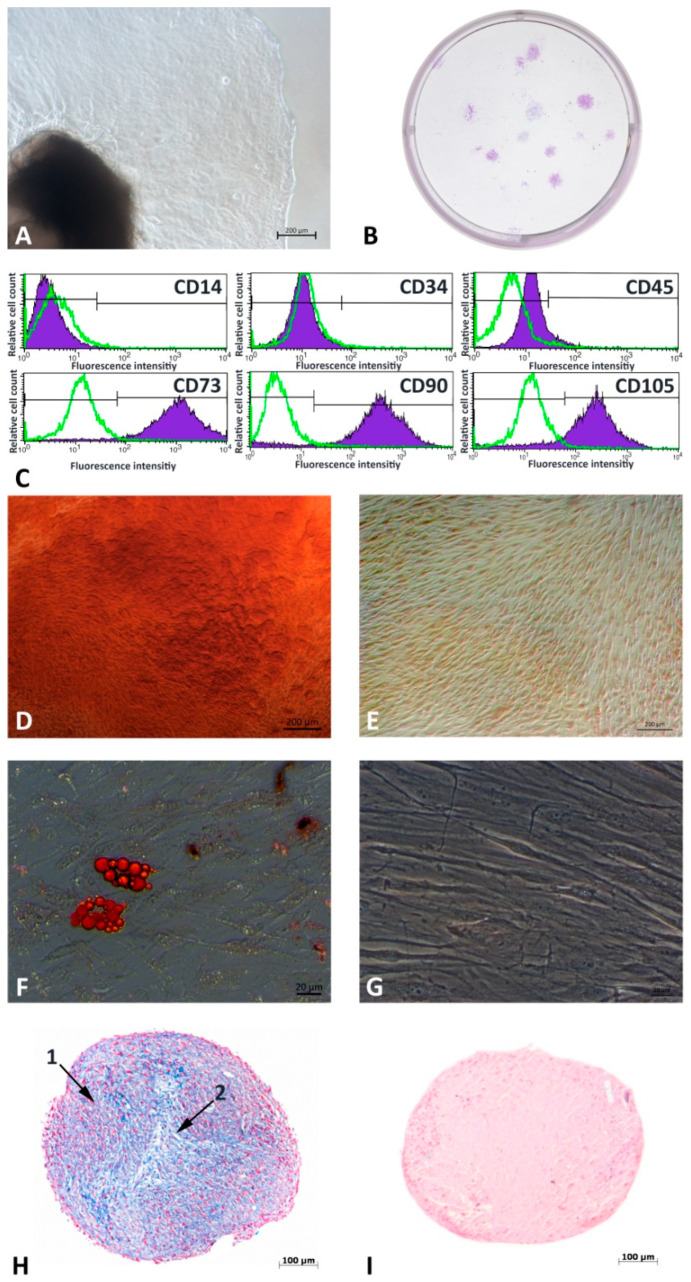
Tissue cultivation and stem cell characteristics. Gingiva sample and outgrowing mixed culture (**A**). Colony formation of MACS^+^ sorted cells after 14 days (**B**). G-MSC specific phenotyping via flow cytometry (**C**). Multilinear differentiation potential (**D**–**I**): Alizarin red staining after osteogenic stimulation (**D**) and control (**E**). Red Oil O staining after adipogenic stimulation (**F**) and control (**G**). Alcian Blue and Nuclear Fast Red stained paraffin-embedded histological samples after 5 weeks of chondrogenic stimulation (**H**) and control (**I**).

**Figure 2 cells-12-02479-f002:**
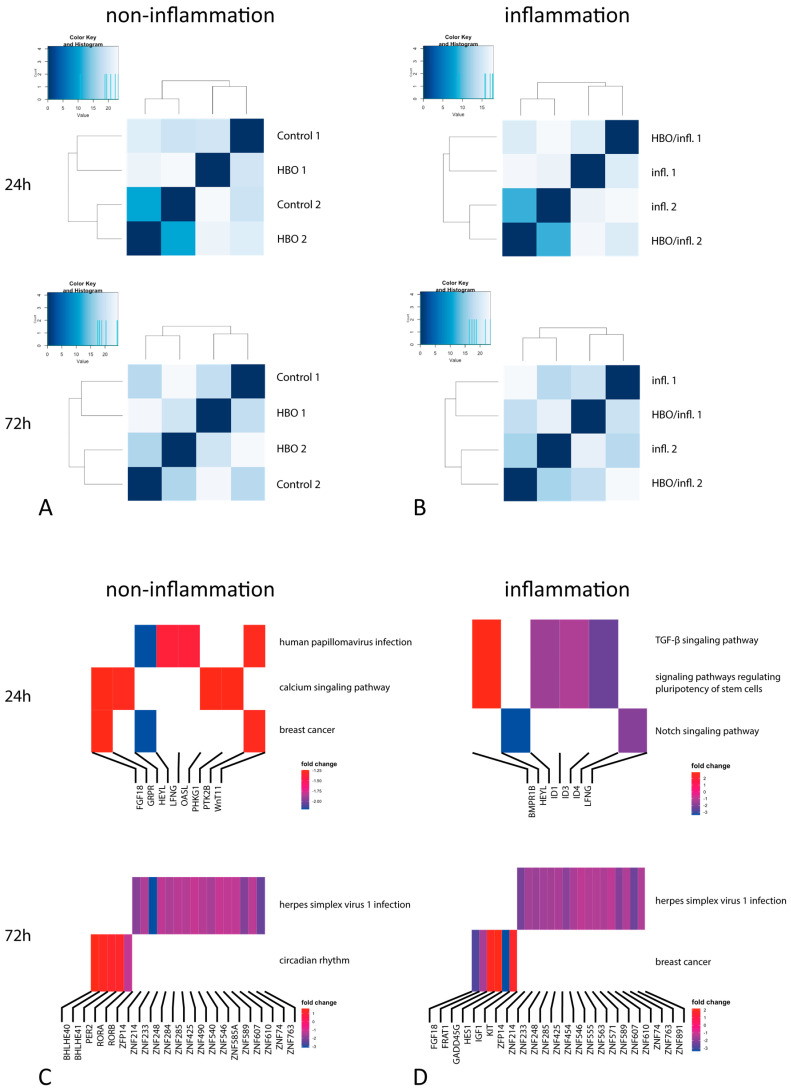
Visualization of gene counts and differentially expressed genes in KEGG pathway analysis. Visualization of gene counts and differentially expressed genes in control-HBO (**A**) and Infl.-inf./HBO (**B**) comparisons: Heatmap of rlog transformed gene counts for all samples following 24 h and 72 h prestimulation. Kyoto Encyclopedia of Genes and Genomes (KEGG) Pathway analysis (**C**,**D**) of all DE genes during 24 h/72 h HBO-stimulation under non-inflammation (**C**) and inflammation conditions (**D**). Fold changes between control and treatment gene expression were highlighted via a color gradient.

**Figure 3 cells-12-02479-f003:**
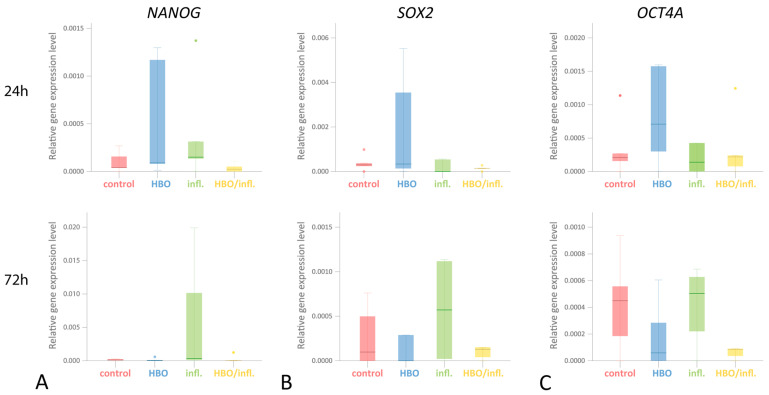
Key transcription factor expression profile under 24 h and 72 h HBO/inflammation conditions. Real-time PCR analysis of *NANOG* (NANOG Homeobox; (**A**)), *SOX2* (sex determining region Y-box 2; (**B**)) and *OCT4A* (octamer-binding transcription factor 4; (**C**)) expression profiles after 24 h and 72 h inflammation/HBO prestimulation. (*n* = 5; box-and-whiskers diagrams with medians and quartiles; outliers are shown as dots).

**Figure 4 cells-12-02479-f004:**
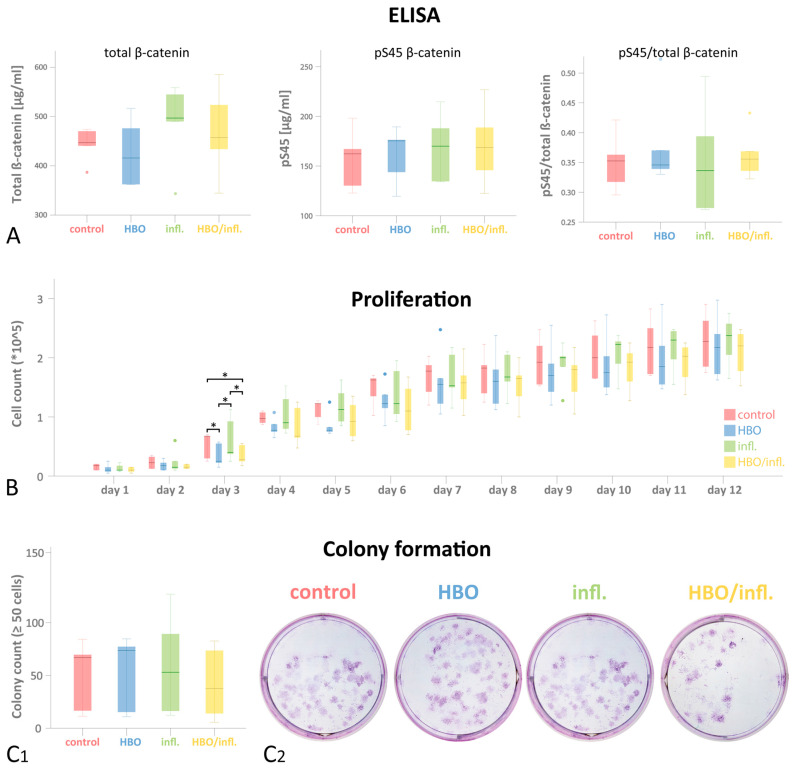
Proliferation assay, colony formation and ELISA after 24 h HBO/inflammation prestimulation. G-MSCs were stimulated over 24 h. pS45- and total β-Catenin expression were analyzed in enzyme-linked immunosorbent assays (**A**). Proliferation assay was observed over 12 days (**B**) and colonies were counted after 12 days (**C1**,**C2**). (*n* = 5; box-and-whiskers diagrams with medians and quartiles; outliers are shown as dots). Significance in Wilcoxon-testing is asterisked (* *p* ≤ 0.05).

**Figure 5 cells-12-02479-f005:**
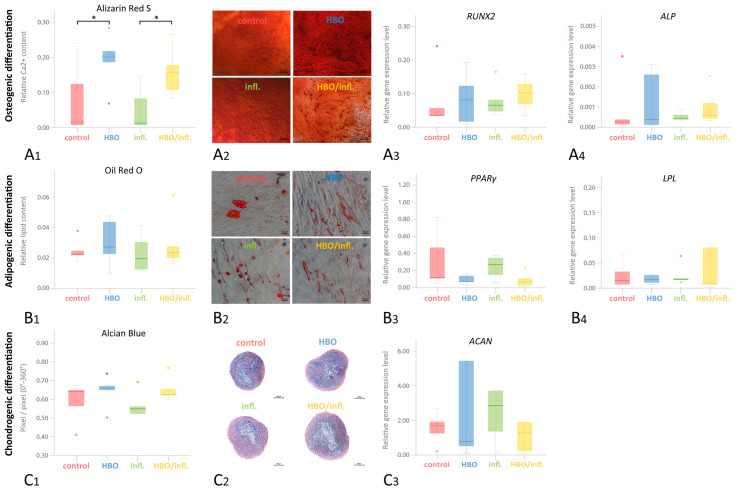
Multilineage differentiation potential under 24 h/72 h HBO/inflammation stimulation. Multilinear differentiation was induced after 24 h inflammation/HBO prestimulation over 3 weeks (osteogenic differentiation; (**A**)), 4 weeks (adipogenic differentiation; (**B**)), and 5 weeks (chondrogenic differentiation; (**C**)). Dyes (**A2**,**B2**) were quantified by elution of Alizarin Red S (**A1**) and Red Oil O (**B1**), and Alcian Blue (**C2**) was quantified in digital image quantification (**C1**). Real-time-qPCR analysis was performed for osteogenic markers *RUNX2* (Runt-related transcription factor 2; (**A3**)) and *ALP* (alkaline phosphatase; (**A4**)), adipogenic markers *PPARγ* (peroxisome proliferator-activated receptor γ; (**B3**)) and *LPL* (lipoprotein lipase; (**B4**)) such as chondrogenic marker *ACAN* (aggrecan; (**C3**)) (*n* = 5; box-and-whiskers diagrams with medians and quartiles, outliers are shown as dots). Significance in Wilcoxon-signed-testing is asterisked (* *p* ≤ 0.05).

**Table 1 cells-12-02479-t001:** Antibodies used for MACS sorting and flow cytometry.

Antibody	Clone	Color	Distributor	Product/Cat. No.
anti-human CD14	REA599	FITC	Miltenyi Biotec (Bergisch Gladbach, Germany)	130-110-576
anti-human CD34	AC136	PE	Miltenyi Biotec	130-113-741
anti-human CD45	REA747	APC	Miltenyi Biotec	130-110-771
anti-human CD73	REA804	PE	Miltenyi Biotec	130-112-060
anti-human CD90	REA897	FITC	Miltenyi Biotec	130-114-901
anti-human CD105	REA794	FITC	Miltenyi Biotec	130-112-327
REA Control (S)	REA293	FITC	Miltenyi Biotec	130-113-437
REA Control (S)	REA293	PE	Miltenyi Biotec	130-113-438
anti-human STRO-1	STRO-1	Alexa Fluor 647	BioLegend (San Diego, CA, USA)	340103
anti-Cy5/Alexa Fluor 647 MicroBeads	-	-	Miltenyi Biotec	130-091-395

**Table 2 cells-12-02479-t002:** Top 3 differentially expressed genes during HBO stimulation under inflammation and non-inflammation conditions.

Contrast	Time	Gene Name	Name	LogFC	*p*-Value	*p*-Value_BH Adjust_
non-inflammation	24 h	*PTK2B*	Protein Tyrosine Kinase 2 Beta	−1.0	2.28 × 10^−17^	3.24 × 10^−13^
		*BMF*	Bcl2 Modifying Factor	−1.0	2.34 × 10^−15^	1.66 × 10^−11^
		*TLL1*	Tolloid Like 1	−1.0	6.72 × 10^−13^	3.19 × 10^−9^
	72 h	*TXNIP*	Thioredoxin Interacting Protein	−2.4	5.57 × 10^−27^	7.90 × 10^−23^
		*PDK4*	Pyruvate Dehydrogenase Kinase 4	−2.5	2.21 × 10^−26^	1.57 × 10^−22^
		*ABL2*	ABL Proto-Oncogene 2, Non-Receptor Tyrosine Kinase	1.7	2.24 × 10^−24^	1.06 × 10^−20^
inflammation	24 h	*PI16*	Peptidase Inhibitor 16	−6.0	8.90 × 10^−22^	1.24 × 10^−17^
		*HMOX1*	Heme Oxygenase 1	−1.0	7.60 × 10^−15^	5.28 × 10^−11^
		*ID3*	Inhibitor Of DNA Binding 3, HLH Protein	−1.0	1.20 × 10^−14^	5.56 × 10^−11^
	72 h	*BHLHE40*	Basic Helix-Loop-Helix Family Member E40	3.0	1.06 × 10^−8^	0.0001
		*ARL4C*	ADP Ribosylation Factor Like GTPase 4C	1.6	5.16 × 10^−7^	0.003
		*HMOX1*	Heme Oxygenase 1	1.7	6.10 × 10^−7^	0.003

Effects have been adjusted for the influence of different probands between control and HBO (non-inflammation) such as infl. and HBO/infl (inflammation) groups. LogFC = log Fold Change, LogCPM = log counts per million. Correction for multiple testing was done with the Benjamini-Hochberg method, level of significance was set to FDR <0.05.

**Table 3 cells-12-02479-t003:** Primers used for RT-qPCR.

Gene	Assay ID	Gene Symbol	Accession ID	Primer Sequence
*PGK-1*	102083	*PGK1* *H. sapiens*	ENST00000373316	FWD: GGAGAACCTCCGCTTTCAT REV: GCTGGCTCGGCTTTAACC
*NANOG*	148147	*NANOG* *H. sapiens*	ENST00000229307	FWD: GAGATGCCTCACACGGAGAC REV: AGGGCTGTCCTGAATAAGCA
*OCT4A*	113034	*OCT4A* *H. sapiens*	ENST00000259915	FWD: GCAAAACCCGGAGGAGTC REV: TCCCAGGGTGATCCTCTTCT
*SOX2*	111867	*SOX2* *H. sapiens*	ENST00000325404	FWD: ATGGGTTCGGTGGTCAAGT REV: GGAGGAAGAGGTAACCACAGG
*RUNX2*	113380	*RUNX2* *H. sapiens*	ENST00000359524	FWD: GCCTAGGCGCATTTCAGAT REV: CTGAGAGTGGAAGGCCAGAG
*ALP*	103448	*ALP* *H. sapiens*	ENST00000374840	FWD: AGAACCCCAAAGGCTTCTTC REV: CTTGGCTTTTCCTTCATGGT
*PPAR* *γ*	110607	*PPARγ* *H. sapiens*	ENST00000287820	FWD: GACAGGAAAGACAACAGACAAATC REV: GGGGTGATGTGTTTGAACTTG
*LPL*	113230	*LPL* *H. sapiens*	ENST00000311322	FWD: TCGTTCTCAGATGCCCTACA REV: GCCTGATTGGTATGGGTTTC
*ACAN*	138057	*ACAN* *H. sapiens*	ENST00000439576	FWD: GAACGACAGGACCATCGAA REV: AAAGTTGTCAGGCTGGTTGG

PGK-1 (phosphoglycerate kinase 1), *NANOG* (NANOG Homeobox), *OCT4A* (octamer-binding transcription factor 4), *SOX2* (sex determining region Y-box 2), *RUNX2* (Runt-related transcription factor 2), *ALP* (alkaline phosphatase), *PPARγ* (peroxisome proliferator-activated receptor γ), *LPL* (lipoprotein lipase), *ACAN* (aggrecan). All primers were supplied by Roche (Mannheim, Germany).

## Data Availability

All data will be available from the corresponding author upon reasonable request.

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
