# Peer review of "Effect of Hyperbaric Oxygen and Inflammation on Human Gingival Mesenchymal Stem/Progenitor Cells"

_cells, 2023, doi:10.3390/cells12202479_

Round 1

Reviewer 1 Report

The manuscript by Tölle et al. provides nest generation seq data on the effect of hyperbaric oxygen application on the characteristics of human gingival MSCs. Treatment with oxygen improves the proliferation, differentiation and stemness of these cells. The effect of inflammation is partially counteracted. The paper is well written, organized and presented. The Figures are informativ.

There are only minor remarks:

1. The acc. number for the platform where the original data sets can be found need to be named before publication.

2. The area of the gingival samples need to be defined in grater detail, Which area of the gingiva was harvested?

3. This referee cant find the patients specification in the manuscript (age, sex, illnesses, smoking status etc.).

4. The antibodies applied, especially the STRO-1 ab, needs to be described in detail.

Reviewer 2 Report

In this manuscript, Johannes Tölle and colleagues investigated the effects of hyperbaric oxygen (HBO) on gingival mesenchymal stem cells’ (G-MSCs) gene expression profile, intracellular pathways activation, pluripotency and differentiation potential under an experimental inflammatory setup, they have shown that there is a positive short-term single HBO anti-inflammatory, regenerative and differentiation stimulatory effect on G-MSCs.

There are several critical issues that the authors must address. I detail my criticisms below:

Major points.

1.      Please deposit the Next Generation Sequencing data and make it public, Provide the dataset number in Materials and Methods.

2.      Where is the result description related to Figures 2A and 2B? Is “Fig. 3A-B” in line 208 should be “Fig. 2A-B”?

3.      The images from Figure 4C2 are totally not clear, which affects the claim of the conclusion.

4.      The figures indicated much more information than those included and written in this manuscript. The authors should dig deeper into these results and improve the writing in all results parts.

Minor points.

1.       There should be spaces between the number and related units from lines 24-27, “G-MSCs were isolated from five healthy individuals (n=5) and characterized.  Single (24h) or double (72h) HBO stimulation (100%O2, 3bar, 90min) was performed under experimental inflammatory [IL-1β(1ng/ml)/TNF-α(10ng/ml)/IFN-γ(100ng/ml)] and non-inflammatory micro-environment”, and other remaining similar places, please amend them.

2.      When mentioning the cell makers such as CD14, CD45 and CD73 in G-MSCs’ Characterization, references should be included.

Round 2

Reviewer 2 Report

Overall the manuscript has significantly improved.

But still need minor revisions in Figures:

In Figure 1F-1G and Figure A2-B2, scale bars should be included in all mentioned images.